# Advancements in Activating Transcription Factor 5 Function in Regulating Cell Stress and Survival

**DOI:** 10.3390/ijms23137129

**Published:** 2022-06-27

**Authors:** Pameila Paerhati, Jing Liu, Zhedong Jin, Tanja Jakoš, Shunyin Zhu, Lan Qian, Jianwei Zhu, Yunsheng Yuan

**Affiliations:** 1Engineering Research Center of Cell & Therapeutic Antibody, Ministry of Education, Shanghai Jiao Tong University College of Pharmacy, Shanghai 200240, China; pmlp129@sjtu.edu.cn (P.P.); liujin058@sjtu.edu.cn (J.L.); jz_daray@sjtu.edu.cn (Z.J.); tanja.jakos@sjtu.edu.cn (T.J.); jianweiz@sjtu.edu.cn (J.Z.); 2Institute of Translational Medicine, Shanghai Jiao Tong University, Shanghai 200240, China; zhushunying@sjtu.edu.cn (S.Z.); qianlan@sjtu.edu.cn (L.Q.)

**Keywords:** ATF5, cell stress, mitochondrial UPR, cell survival

## Abstract

Activating transcription factor 5 (ATF5) belongs to the activating transcription factor/cyclic adenosine monophosphate (cAMP) response element-binding protein family of basic region leucine zipper transcription factors. ATF5 plays an important role in cell stress regulation and is involved in cell differentiation and survival, as well as centrosome maintenance and development. Accumulating evidence demonstrates that ATF5 plays an oncogenic role in cancer by regulating gene expressions involved in tumorigenesis and tumor survival. Recent studies have indicated that ATF5 may also modify the gene expressions involved in other diseases. This review explores in detail the regulation of ATF5 expression and signaling pathways and elucidates the role of ATF5 in cancer biology. Furthermore, an overview of putative therapeutic strategies that can be used for restoring aberrant ATF5 activity in different cancer types is provided.

## 1. Introduction

Activating transcription factor 5 (ATF5) is a member of the activating transcription factor/cyclic adenosine monophosphate (cAMP) response element-binding protein (ATF/CREB) family of basic region leucine zipper (bZIP) transcription factors [1]. At first, an ATF5 mouse homolog was discovered and isolated from mouse cDNA expression libraries using protein blot analysis and referred to as an activating transcription factor 4 (ATF4)-related ATFx [2]. Later, the isolation of human ATF5 was also performed from a cDNA library in a yeast two-hybrid screen of human cell division cycle 34 (hCdc34)-interacting genes and named hATF5 [3]. Additionally, activating transcription factor 7 (ATF7) was initially described as a truncated form of ATF5, but later studies revealed that it is structurally and functionally more closely related to the activating transcription factor 2 (ATF2) family [4,5].

Transcription factors can activate and repress the transcription of genes, which are involved in many biological functions, including cell proliferation, differentiation, homeostasis maintenance, and cancer development. As a member of the ATF family, ATF5 is involved in many cellular biological processes. ATF5 inhibits apoptosis caused by cytokine deprivation in an interleukin 3 (IL-3)-dependent cell line, indicating its anti-apoptotic role in cell survival [6]. ATF5 also has a critical role in regulating tissue development, cellular differentiation, and stress responses, and has a promotive role in cancer survival [7,8]. Furthermore, recent studies have also revealed the protective role of ATF5 in mammalian mitochondrial dysfunction and its relation to cancer development [9].

Many new findings have been noted regarding the role of ATF5 in biological functions, post-translational modification, and up- or down-stream genes in both in vitro and in vivo studies. These findings bring new insights into the mechanisms by which ATF5 is involved in cancer, endoplasmic reticulum (ER) stress, and mitochondrial dysfunction.

## 2. Structure and Post-Translational ATF5 Modification

### 2.1. Molecular ATF5 Properties

Like other ATF/CREB family proteins, ATF5 has a bZIP domain, which consists of the basic N-terminal DNA-binding domain that binds to the ATF/cAMP response element (CRE) binding site consensus sequence TGACGT(C/A) (G/A), and the C-terminal leucine zipper region that directs homo- and hetero-dimerization (Figure 1) [1,10]. The bZIP domain is highly conserved in the ATF/CREB family, and the classification of ATF/CREB family members is based on the similarity of their bZIP domain amino acid sequences and dimerization properties [10,11]. ATF5 has been classified into the ATF4 subgroup family, as a protein sequence comparison showed that both human and mouse ATF5 bZIP domains share more than 55% similarity with ATF4 [7,11]. Over-expression experiments on dominant-negative (d/n) ATF5 revealed that ATF5 can form a heterodimer with ATF4 and CCAAT/enhancer-binding protein (C/EBP) in cells [12]. Two distinct ATF5 mRNA variants, ATF5α and ATF5β, have been confirmed in both humans and mice, but both mRNAs code the same protein [11,13].

The ATF5 gene appears to have been structurally and functionally conserved during evolution, because its bZIP domain shares high sequence similarity across many different species. Additionally, the full length of a mouse Atf5 features an amino sequence that is approximately 87% identical to human ATF5 [11]. Furthermore, the amino acid sequence of the zebrafish Atf5’s bZIP domain is more than 72% similar to medaka, frog, mouse, and human Atf5 bZIP [14].

### 2.2. Post-Translational ATF5 Modification

Protein post-translational modification widely exists in mammalian proteins. Tyrosine phosphorylation in mouse ATF5 was first described in an in vitro study in which the pleiotropic regulatory locus 1 (PRL1) protein tyrosine phosphatase was directly bound to the ATF5 bZIP domain and dephosphorylated [5]. Owing to advancements in phosphoproteomics, three phosphorylated sites, i.e., serine-220 (S220), tyrosine-225 (Y225), and serine-256 (S256), were confirmed with a liquid chromatography–tandem mass spectrometry analysis [15]. However, their biological functions are still unclear. A recent study purported that ATF5 could be phosphorylated and stabilized by mitogen-activated protein kinase (MAPK)-related nemo-like kinase (NLK), probably at serine-92 (S92), L-threonine-94 (T94), serine-126 (S126), and serine-190 (S190) (Figure 1) [16]. This phosphorylation is triggered by interleukin 1β (IL-1β)-induced transforming growth factor β-activated kinase 1 (TAK1)/NLK phosphorylation and selectively controls ATF5 function, possibly by influencing its interactions with binding partners (e.g., C/EBP) or other ATF5 modifications; nevertheless, its exact mechanisms require further elucidation.

The N-terminal amino acids 1–21 of ATF5 are required for IL-1β-induced increased ATF5 translation efficiency by protein kinase R(PKR)-like ER kinase (PERK)-mediated eukaryotic translation initiation factor 2a (eIF2a) phosphorylation in a c-Jun N-terminal kinase (JNK)-dependent manner (Figure 1) [17]. However, a previous study indicated that ATF5 phosphorylation may not be required for transcriptional activation, because it contains a constitutively active transcriptional activation domain [12].

ATF5 is acetylated at lysine-29 (K29) by interacting with the transcriptional coactivator e1A-binding protein p300 (P300) (Figure 1), which in turn enhances the interaction between ATF5 and P300 and activates the early growth response 1 (EGR1) gene, thereby regulating tumorigenesis and tumor progression in glioblastoma and breast cancer [18,19]. Another study reported that ATF5 can directly interact with C/EBPβ through the P300-dependent acetylation of ATF5 and bind to the C/EBPα promoter, thereby enhancing the C/EBPβ transactivation of C/EBPα, which is a regulator of adipocyte differentiation [20]. Moreover, ATF5 acetylation at K29 in a P300-dependent manner by immediate-early 2 gene product (IE86) human cytomegalovirus (HCMV) (86 kDa immediate-early protein) is known to promote glioma cell survival [21].

In addition, ATF5 must be ubiquitinated by CDC34 and RAD6B to be degraded by the ubiquitin–proteasome system [3]. Interestingly, the chemotherapy drug cisplatin can inhibit ATF5 ubiquitination and improve ATF5 stability in cells by blocking the interaction between ATF5 and CDC34 [22]. The mechanism of ATF5 ubiquitination is independent of lysine residues and takes place at the N-terminal methionine residue. The ubiquitin–proteasome degradation of ATF5 can also be down-regulated by cadmium and NLK phosphorylation, increasing the ATF5 protein stabilization [16,23]. Unlike ubiquitination, the ATF5 small ubiquitin-like modifier (SUMO)ylation takes place at lysine residues of the protein. Both lysine-106 (K106) and lysine-107 (K107) (Figure 1) can be conjugated by (SUMO) 2/3 [24]. ATF5 SUMOylation or deSUMOylation is involved in regulating the ATF5 localization on the centrosome [25].

## 3. Biological ATF5 Functions

### 3.1. Cell Differentiation and Tissue Development

ATF5 regulates stem cell differentiation and is essential for tissue development. In the development of cerebral neurons, ATF5 inhibits the differentiation and maintains the proliferative state of neural progenitor cells into neurons, astrocytes, and oligodendroglia [26,27,28]. ATF5 also regulates cerebellar granule neuron development by increasing the sonic hedgehog protein-mediated proliferation and neuronal differentiation [29]. Several studies on ATF5 knockout mice revealed that ATF5 deficiency leads to neonatal death, olfactory defects, olfactory bulb hypogenesis, behavioral abnormalities, and the abnormal development of the cerebral cortex [30,31,32,33]. In addition to neural development, ATF5 is involved in the regulation of non-neural differentiation. ATF5 mediates the promotion of adipocyte differentiation, hepatocyte differentiation, and the inhibition of osteogenic differentiation, [20,34,35]. A recent study revealed that ATF5 is enriched in odontoblast-related nucleosome-free regions and promotes odontoblast terminal differentiation by increasing the enhancer activity of dentin matrix protein 1 (DMP1), which is mainly expressed in the odontoblast layer and important for bone development [36]. Another study demonstrated that the knockdown of ATF5 in murine erythroid progenitor cells significantly reduced the proliferation of fetal liver erythroid progenitors and inhibited erythroid differentiation [37].

### 3.2. Centrosome Maturation

ATF5 acts as a structural protein that regulates and maintains centrosome integrity. During the centrosome duplication cycle, ATF5 regulates the pericentriolar material-dependent centriole formation by interacting with pericentrin and polyglutamylated tubulin [25]. ATF5 SUMOylation occurs in a cell-cycle-dependent manner, and the highest and lowest levels of SUMOylation take place at the G1 and M phases, respectively. The SUMOylation of ATF5 interferes with its association with centrosome proteins at the molecular level. The mutation of SUMOylation sites in ATF5 can improve its accumulation on the centrosomes at the G1 phase and block its dissociation of centrosomes [24]. The centrosome cycle changes usually result in centrosome abnormalities that induce genome instability and tumorigenesis. This mechanism can partially explain ATF5’s regulation of centrosome maturation and its high expression in many types of tumor cells.

### 3.3. Responses to Cellular Stress

ATF5 participates in the cellular stress response to ER stress, mitochondria dysfunction, arsenite exposure, and proteasome inhibition [38,39]. Abundant evidence exists proving that ATF5 is involved in pro-survival pathways activated under cellular stress, such as the ER unfolded protein response (UPR), mitochondrial UPR (UPRmt), and heat-shock response (HSR). ATF5 over-expression can protect neural, retinal ganglion, chondrocyte, and pancreatic β cells from apoptosis under ER stress [40,41,42,43]. ATF5 also promotes cardiomyocyte survival by activating heat-shock protein 27 (HSP27) under hyperthermia-induced heat-shock stress in cardiomyocytes [44]. Meanwhile, up-regulated levels of ATF5 in mitochondrial dysfunction promote mitochondrial oxidative phosphorylation (OXPHOS) recovery and induce UPRmt by mediating the transcription of mitochondrial chaperones (heat-shock protein 60 (HSP60) and mitochondrial heat-shock protein 70 (mtHSP70)) and proteases (mitochondrial Lon peptidase 1 (LONP1)) to promote survival and recovery through maintaining protein homeostasis under mitochondrial stress [9,45]. Among these, HSP60 and mtHSP70 conjointly promote the correct folding of denatured and newly synthesized polypeptides. LONP1 enhances the degradation of irreversibly damaged proteins by cleaving them and mediates the remodeling of OXPHOS complexes, which are needed for the survival and proliferation of tumor cells [46,47,48].

The protective role of ATF5 in cardiac mitochondrial dysfunction has become clearer with increasing research on the function of ATF5-regulated UPRmt. ATF5 activates UPRmt-induced protection in cardiac ischemia–reperfusion injury (I/R), and the global knockdown of ATF5 results in the failure of the cardioprotective effects of the UPRmt inducers oligomycin or doxycycline in I/R [49]. ATF5 also participates in the activation of the UPRmt-mediated cardioprotective role of tetrahydrocurcumin (THC), a traditional treatment for cardiovascular disease, and in pathological cardiac hypertrophy by forming a signaling axis with peroxisome proliferator-activated receptor-gamma co-activator-1 α (PGC-1α), which is a regulator of mitochondrial quality control and energy metabolism [50]. The aberrant expression of ATF5 has emerged as one of the biomarkers for increased UPRmt activity and improved mitochondrial function. For instance, the significant up-regulation of ATF5 mRNA and protein levels might reflect the UPRmt-enhanced mitochondrial resistance to early cocaine-induced cardiotoxicity. Additionally, a decreased ATF5 expression indicates UPRmt inactivation and insufficient mitochondrial recovery in heart failure with a preserved ejection fraction (HFpEF) [51,52].

## 4. Transcriptional Regulation and Down-Stream Targets of ATF5

ATF5 expression is regulated at both the transcriptional and translational levels in response to different stresses, subsequently affecting the activation of ATF5 down-stream targets and the related signaling pathways that are involved in cell survival. The translational ATF5 expression in response to environmental stress is regulated by eIF2a phosphorylation, which is in turn regulated by the kinases serine/threonine-protein kinase (GCN2), PERK, or PKR [38]. Along with ATF5, eIF2a enhances ATF4 translation, which is necessary for increased ATF5 transcription. In addition, several other up-stream transcriptional factors can activate ATF5 expression, thereby promoting the expression of pro- and anti-survival target genes that regulate the cell survival pathways in cancer development and during cellular stress, inflammation, and metabolic maintenance.

### 4.1. Survival Pathways in Cancer Development

Accumulating evidence indicates that ATF5 regulates tumor survival and development by mediating a myriad of survival pathways, such as apoptosis, autophagy, tumorigenesis, migration, and invasion (Figure 2). In the anti-apoptotic and anti-autophagy pathways, ATF5 transcription is enhanced by cAMP-responsive element-binding protein 3-like 2 (CREB3L2), whose expression is up-regulated by the RAS/MAPK and phosphatidylinositol 3-kinase (PI3K)/protein kinase B (AKT) signaling cascades, which are activated by fibroblast growth factor receptor substrate 2 (FRS2) and p21-activated kinase 1 (PAK1) [53]. ATF5 then activates the transcription of the anti-apoptotic gene myeloid cell leukemia sequence 1 (MCL1) transcription by binding its promoter and increasing the malignant glioma survival through the CREB3L2/ATF5/MCL1 anti-apoptotic pathway. ATF5 can also activate another anti-apoptotic gene, B cell leukemia/lymphoma 2 (BCL2) transcription, therefore, increasing glioblastoma and breast cancer survival. The oncoprotein BCR-ABL activates the PI3K/AKT signaling cascade, which inhibits the ATF5 transcriptional repressor forkhead box O4 (FOXO4) transcription to increase ATF5 expression. Then, the transcription of the anti-autophagy gene mammalian rapamycin target (mTOR) is triggered, resulting in the suppression of autophagy in BCR-ABL-transformed myeloid leukemia cells [54]. Moreover, ATF5 is a down-stream effector of protein arginine methyltransferase 1 (PRMT1), which is an essential regulator of neuroblastoma cell survival [55]. ATF5 is significantly down-regulated in PRMT1-depleted cells, and the over-expression of ATF5 can rescue the neuroblastoma cells from PRMT1 depletion-induced apoptosis.

In the tumorigenesis and migration pathways, ATF5 transcription can be activated by the tumor cell growth and migration promoter E74-like E26 transforming sequence (ETS) transcription factor (ELF1), increasing the malignancy of gliomas [56]. ATF5 directly activates the transcription of the disheveled segment polarity protein gene (DVL1), an essential regulator of the Wnt/β-catenin down-stream effectors (e.g., proto-oncogenes basic helix–loop–helix (bHLH) transcription factor MYC and activating protein (AP-1) transcription factor subunit JUN, cell cycle control gene cyclin D1, as well as non-kinase transmembrane proteoglycan gene CD44), which then mediates the Wnt/β-catenin pathway in bladder cancer and promotes tumorigenesis [57]. ATF5 also increases the hypoxia-inducible factor 1 (HIF1) transcriptional complex activity by binding to HIF1α and promoting the transcription of HIF1 target genes, including 3-phosphoinositide-dependent protein kinase gene (PDK1), carbonic anhydrase (CA9), plasminogen activator inhibitor (PAI1), and vascular endothelial growth factor (VEGFA) genes. This further activates HIF1-mediated tumorigenic signaling and tumor angiogenesis, which can be inhibited by d/nATF5 in vivo esophageal cancer models [58]. Furthermore, ATF5 initiates expression of its pro-invasion down-stream targets, the collagen receptor integrin α2β1 genes ITGα2 and ITGβ1, which can bind to collagen type I, E-cadherin, and matrix metalloproteinase to induce tumor invasion in several human cancer cell lines, such as lung, breast, cervical, gastric, fibrosarcoma, and pancreatic cancers [59,60].

ATF5 can stimulate the proapoptotic pathways in cancer (Figure 2). It can bind to the inhibitor of DNA-binding helix–loop–helix (HLH) protein gene (ID1) promoters at the CRE site and inactivate ID1 transcription, repressing cell proliferation and leading to a G2-M cell cycle arrest during hepatocellular carcinoma (HCC) development [61]. ATF5 can also bind to the cyclin D3 gene promoter and increase its transcription via an interaction with early two-factor family transcription factor 1 (E2F1), resulting in increased cisplatin-induced apoptosis in human cervical cancer cells [62]. ATF5 and cyclin D3 interact in those tumor cells through a positive feedback loop in which cyclin D3 enhances the ATF5 transcription [63].

### 4.2. Survival Pathways during Cell Stress and Inflammation

ATF5 transcription is also associated with pro- and anti-survival pathways during ER, mitochondrial, and other types of cellular stresses (Figure 3). ATF5 represses ER stress-induced apoptosis through the BBF2H7 (CREB3L2)/ATF5/MCL1 anti-apoptotic pathway in chondrocyte differentiation [43]. Conversely, ATF5 maintains protein homeostasis by co-stimulating BCL2 homology 3 (BH3)-only proapoptotic protein (NOXA)-regulated proapoptotic pathways with ATF4 and DNA damage-inducible transcript 3 (CHOP) [64]. During different stress conditions (e.g., ER unfolded protein, nutrient deprivation, and oxidative damage), the translation of ATF4 and CHOP is enhanced by the phosphorylation of eIF2, and then both bind to the ATF5 promoter and enhance ATF5 transcription. Then, CHOP and ATF5 jointly stimulate the NOXA, thus, resulting in increased cell apoptosis in order to maintain protein homeostasis. ATF5 is also activated by mitochondrial nuclear retrograde regulator 1 (MNRR1), a regulator of mitochondrial metabolism, and induces mitophagy and autophagy to maintain mitochondrial homeostasis during ER stress-induced UPRmt in a MNRR1/ATF5-dependent manner [65].

ATF5 might be involved in the developmental pathway of diabetes by regulating pancreatic β cell survival during cellular stresses and metabolic homeostasis. During ER stress, the human diabetes gene pancreatic and duodenal homeobox 1 (PDX1)-induced ATF4 and ATF5 activation restores the cellular homeostasis and the survival of pancreatic β cells by activating the expression of the protein translation suppressor gene eukaryotic translation initiation factor 4E-binding protein 1 (4EBP1) through the PDX1–ATF transcriptional complex [42,66]. On the other hand, ATF5 mediates β cell apoptosis by activating the transcription of the proapoptotic glutathione-degrading enzymes cation transport regulator 1 (CHAC1) and glutamate pyruvate transaminase 2 (GPT2), also through the PDX1–ATF axis, by up-regulating the expression of the thioredoxin interacting protein (TXNIP), which can trigger the inflammasome into producing proinflammatory cytokines (e.g., IL-1β) and inducing inflammation under ER stress [66,67]. Furthermore, ATF5 also participates in insulin resistance and increases lipogenesis through the β-catenin signal activator compound 21 (CP21)-induced Wnt/β-catenin pathway in hepatocytes, as well as influences chronic inflammation and lipid metabolism by down-regulating the expression of serum amyloid genes SAA1 and SAA2, which are overproduced in response to IL-1β-induced inflammatory stress [17,68].

Additionally, ATF5 can promote metabolic maintenance. ATF5’s transcription is stimulated by the mitochondrial fission 1 (FIS1)-triggered accumulation of the tricarboxylic acid cycle (TCA) intermediate under high-fat diet-induced mitochondrial stress in liver cells [69]. The over-expression of ATF5 causes the inhibited transcription of interferon regulatory factor 3 (IRF3), a regulator of type I interferons (IFN-Is), including interferon alpha (IFNα) and beta (IFNβ), further increasing the mitophagy activity and suppressing the IFN-I-mediated metabolic inflammation pathway. Meanwhile, ATF5 is translationally expressed by GCN2-dependent eIF2α phosphorylation and regulates metabolism, cell growth with ATF4 by conjointly activating fibroblast growth factor 21 (FGF21), and a pressure sensor in the liver that regulates metabolism via binding to its promoter amino acid response elements site during periods of nutrient restriction [70]. All indicate that ATF5 participates in the restoration of hepatic metabolic homeostasis in liver cells.

### 4.3. Reverse ATF5 Regulation

In addition to factors that enhance the transcriptional activities of ATF5, several known repressors of ATF5 exist (Figure 4). The ATF5–nucleophosmin 1 (NPM1) protein interaction promotes ATF5 degradation through proteasome- and caspase-dependent pathways in HCC cells. This replaces the interaction of the ATF5 protein with heat-shock protein 70 (HSP70), which maintains high levels of ATF5 expression by preventing its degradation in glioma cells [71,72].

ATF5 post-transcriptional expression can also be suppressed by miRNAs, including miR-141-3p, miR-520b-3p, and miR-134-5p, which bind to the 3′ UTR site of ATF5 mRNA and regulate ATF5 expression. In this process, the down-regulation of ATF5 expression by miR-141-3p inhibits cell proliferation and promotes cell apoptosis in gliomas [73]. miR-520b-3p can suppress ATF5 expression levels and play a positive role in tumor inhibition during diverse stress conditions in ATF5-oncogenic cancer cells; however, the specific mechanisms of miR-520b-3p-induced ATF5 inhibition are not fully understood and require further investigation [74]. Furthermore, HCMV infection can trigger miR-134-5p down-regulation, which is the negative regulator of ATF5 expression, thus, resulting in high levels of ATF5 and increased glioma cell survival [75].

## 5. ATF5 in Cancer Therapy

ATF5 is highly expressed in most cancers, including myeloid leukemia, glioma, breast, ovarian, lung, colorectal, gastric, rectal, and pancreatic cancer [76,77,78]. In these cancers, ATF5 is an attractive target for cancer therapy because of its crucial role in cell differentiation, tissue development, and cellular stress responses that promote cancer survival. Additionally, recent studies have revealed that ATF5 is a potential therapeutic target for many other cancers. A study that screened for potential diagnostic and therapeutic markers for osteosarcoma revealed that ATF5 is among the up-regulated genes that are strongly associated with poor prognosis and recurrence, indicating that ATF5 can be a putative target for osteosarcoma treatment [79]. ATF5 is also essential for promoting tumor growth in mammary tumor cells in which ATF5 knockdown reduces the proliferation and migration of CD24^+^ Mvt1 cells [80]. Moreover, ATF5 promotes tumorigenesis and has prognostic and therapeutic value in esophageal cancer and urothelial bladder cancer [57,58].

At first, a dominant-negative form of the ATF5 protein (d/nATF5) was designed to investigate the role of ATF5 in neural differentiation, which is structurally modified in the N-terminal DNA-binding domain to interfere with ATF5 functions [27,81]. However, subsequent studies showed that d/nATF5 can promote apoptotic cell death and lead to tumor regression in several cancer cells, both in vitro and in vivo, but does not affect normal cells [82]. These results indicate that d/n ATF5 blocks ATF5 function in cancer cells and has therapeutic potential for targeting ATF5-overexpressing tumors. After these findings, different forms of d/n ATF5, such as cell-penetrating d/n-ATF5-recombinant peptide (CP-d/n-ATF5-RP) and cell-penetrating d/n-ATF5-synthesized peptide (CP-d/n-ATF5-S1), were designed and investigated in various cancer types with promising results. Both peptides are effective and non-toxic to normal tissue, and their effect is not only limited to glioma cells, but was also shown in melanoma and prostate and breast cancer [83,84]. CP-d/n-ATF5-S1 triggers the proapoptotic mechanism by repressing BCL2 and MCL1 expression through a reduced expression of deubiquitinase ubiquitin-specific peptidase 9 X-linked (Usp9X) [83]. A recent study reported that CP-DN-ATF5 can also interfere with endogenous CCAAT enhancer-binding protein beta (CEBPB) and delta (CEBPD) activities, whose inhibition promotes tumor cell death [85]. Consequently, Bpep and Dpep were designed to interfere with ATF5, CEBPB, and CEBPD and promote cell death by down-regulating the anti-apoptotic protein survivin and up-regulating proapoptotic BCL2-modifying factor (BMF) [86]. CP-DN-ATF5 has also been shown to induce apoptosis by down-regulating survivin [87].

In addition to antagonist treatment, a nanoplatform, siRNA-CaP-rHDL (ATF5 siRNA loaded by calcium phosphate into high-density lipoprotein apolipoprotein E3 nanoparticles), was built to deliver ATF5 siRNA safely and efficiently in glioblastoma cells, both in vitro and vivo, providing new insights into ATF5-targeted treatments [88].

## 6. Future Directions and Conclusions

Abnormal ATF5 expression that is induced by various up-stream factors results in the activation of its oncogenic down-stream pathways, including survival factors that mediate apoptosis, cell growth, autophagy, tumor migration factors, and mitochondrial protection genes, which together promote tumor survival and migration. Likewise, ATF5 activates targets that regulate cellular inflammation, homeostasis, and metabolism. An enhanced understanding of these signaling networks could provide a new direction for the development of therapeutic options that target or influence ATF5 activity. Some examples include inhibiting PRMT1/ATF5 in neuroblastomas, increasing the level of ATF5 inhibition-related miRNAs in gliomas, targeting the Fis-1/ATF5 axis in metabolic diseases, or targeting ATF5 in diabetes. Recent studies have also revealed that ATF5 increases the radiation resistance of cancer cells by promoting cell cycle progression, cell growth, and invasiveness, indicating that it may reduce the effects of radiation therapy; perhaps using ATF5 antagonists may prevent this [89,90]. However, these potential treatments still require deeper research into ATF5 target genes and their functions in diseases.

In addition to ATF5’s carcinogenic role, it appears to suppress tumor growth in HCC. Generally, ATF5 is highly expressed in normal liver cells and is essential for liver development [91,92]. However, in HCC, ATF5 is significantly down-regulated via its promoter methylation, and a recovered ATF5 level can inhibit HCC cells growth; the suppression of ATF5 by promoter hypermethylation could also enhance cell growth in HCC cell lines. Moreover, low levels of ATF5 have been associated with increased tumor malignancy and can be used as a prognostic indicator in HCC [61,93,94]. Nevertheless, underlying mechanisms governing the inhibitory effects of ATF5 in HCC are still unclear, and the lack of in vivo evidence suggests that this may be a relevant issue to be explored in future studies.

Taken together, ATF5 is a prospective target for cancer therapy and a prognostic biomarker that has various functions, including promoting tumorigenesis, tumor invasion, and radioresistance. ATF5 is also involved in metabolic maintenance, insulin resistance, and chronic inflammation. These all indicate the therapeutic potential of ATF5 in regulating a variety of diseases, and more in-depth research may lead to multiple therapeutic options. Therefore, the discovery of complex ATF5 functions remains an active area of study.

## Figures and Tables

**Figure 1 ijms-23-07129-f001:**
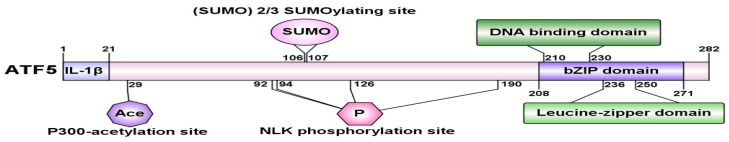
Protein structure of activating transcription factor 5 (ATF5) (NP_001180575). Human ATF5 has 282 amino acids (aa). Long protein of 30 kDa that consists of the N-terminal (1–21 aa) interleukin 1 β (IL-1β) immune response domain, E1A-binding protein p300 (P300)-acetylation lysine-29 (K29), nemo-like kinase (NLK), phosphorylation serine-92 (S92), L-threonine-94 (T94), serine-126 (S126), serine-190 (S190), and small ubiquitin-like modifiers (SUMO) 2/3 SUMOylation lysine-106/lysine-107 (K106/K107) sites. The C-terminal bZIP domain (208–271 aa) contains a DNA-binding domain (210–230 aa) and leucine zipper domain (236–250 aa). *Ace*, acetylation; *P*, phosphorylation; *SUMO*, SUMOylation.

**Figure 2 ijms-23-07129-f002:**
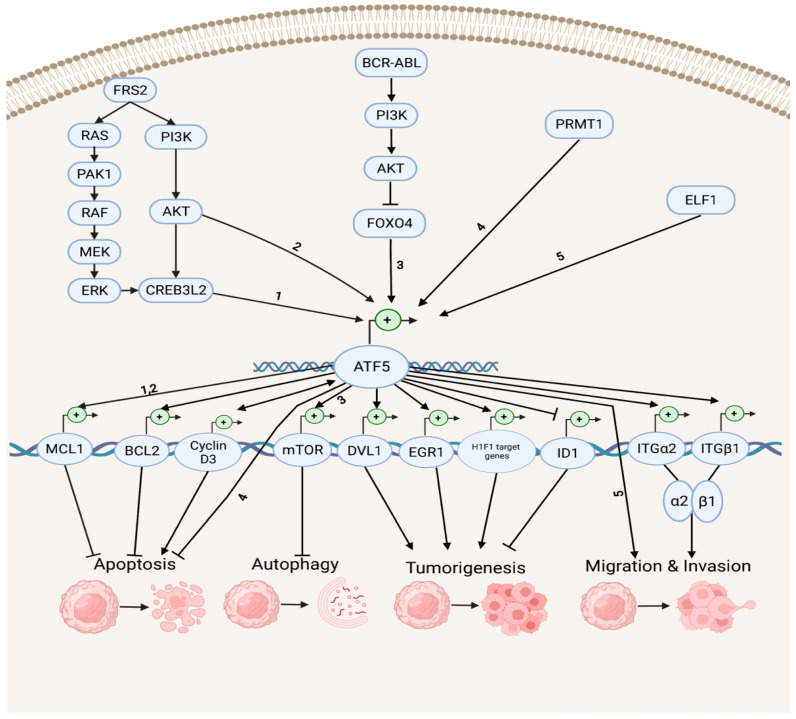
Survival pathways in cancer development. Depicted are up-stream activators of ATF5 and its down-stream target genes that regulate diverse cellular processes, including anti-apoptosis, apoptosis, anti-autophagy, tumorigenesis, migration, and invasion. The numbers represent the known connections between related elements of the signaling pathways.

**Figure 3 ijms-23-07129-f003:**
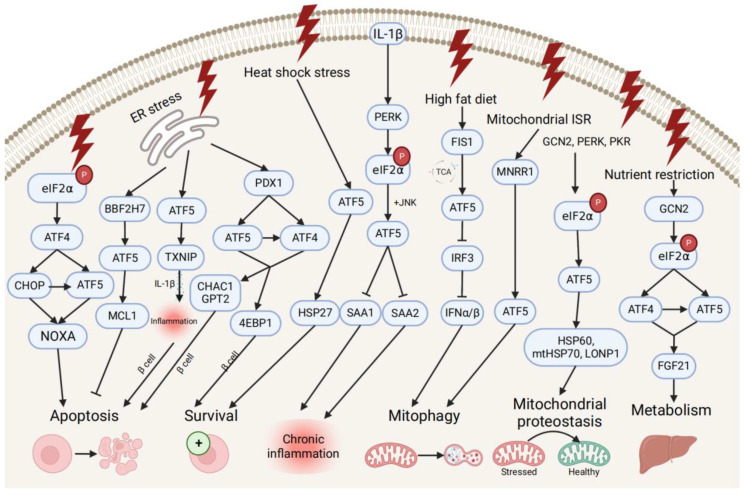
Pro- and anti-survival pathways during cellular stress and inflammation. The up-stream activators of ATF5 are shown, along with target genes that respond to diverse cellular stressors through the induction of anti-apoptotic or apoptotic signals and pathways that promote cell survival, regulate metabolism, chronic inflammation, and mitophagy, and maintain mitochondrial proteostasis.

**Figure 4 ijms-23-07129-f004:**
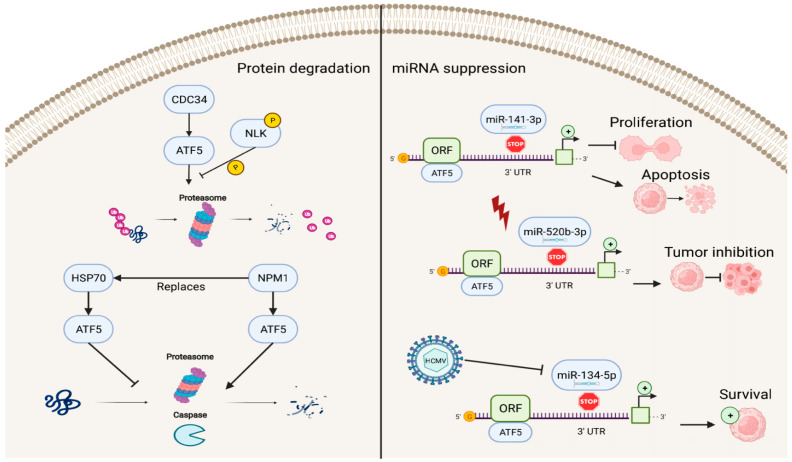
Reverse regulation of ATF5 expression. ATF5 can be degraded by cell division cycle 34 (CDC34), nucleophosmin 1 (NPM1), and miRNAs (miR-141-3p, miR-520b-3p, and miR-134-5p), resulting in inhibition of cell proliferation, induction of apoptosis, and decreased survival of cancer cells.

## Data Availability

Not applicable.

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
