# Peer review of "Advancements in Activating Transcription Factor 5 Function in Regulating Cell Stress and Survival"

_ijms, 2022, doi:10.3390/ijms23137129_

Round 1

Reviewer 1 Report

Dear All,

The authors have completed a thorough review of the role of activating transcription factor 5 (ATF5) in cell stress, differentiation, and survival. In addition, the review included a discussion on centrosome maintenance and, most significantly, the function of ATF5 governing tumorigenesis and the survival of tumor cells. Finally, the review included the involvement of ATF5 in diabetes and the maintenance of metabolism.

The manuscript is acceptable after minor revisions are accomplished.

1)  The manuscript mentions ATF5 as an oncogene. However, there are no publications that indicate that ATF5 induces tumors. Instead, ATF5 participates in oncogenesis. Therefore, the authors need to add supporting evidence that ATF5 is indeed an oncogene or focus on ATF5 participation in oncogenesis/tumorigenesis.  

2) Text lines 91-97 are unclear whether ubiquitination stabilizes ATF5 as a functional transcription factor or stabilized in a nonfunctional state before degradation.

Author Response

Reviewer1# comments

The authors have completed a thorough review of the role of activating transcription factor 5 (ATF5) in cell stress, differentiation, and survival. In addition, the review included a discussion on centrosome maintenance and, most significantly, the function of ATF5 governing tumorigenesis and the survival of tumor cells. Finally, the review included the involvement of ATF5 in diabetes and the maintenance of metabolism.

The manuscript is acceptable after minor revisions are accomplished.

1)The manuscript mentions ATF5 as an oncogene. However, there are no publications that indicate that ATF5 induces tumors. Instead, ATF5 participates in oncogenesis. Therefore, the authors need to add supporting evidence that ATF5 is indeed an oncogene or focus on ATF5 participation in oncogenesis/tumorigenesis.  

Thank you for your correction. We initially wanted to express it to promote tumorigenesis, not to induce tumors. We have corrected all mistakes related to oncogenes in the manuscript to “ ATF5 participation in tumorigenesis”. Page 1, line 39-40; page 9, line 341.

2)Text lines 91-97 are unclear whether ubiquitination stabilizes ATF5 as a functional transcription factor or stabilized in a nonfunctional state before degradation.

We checked our manuscript and reference in detail. This mechanism mentioned in the literature is from author's hypothesis,but they have no more evidence to support their theory. We have deleted it in the new version of manuscript.

Other changes:

  1. We checked spelling of author’s names and corrected two typos in this version of manuscript.
  2. In page 3, line 97 “N-terminal amino acids 1-20 of ATF5” should be “N-terminal amino acids 1-21 of ATF5”, and “directly interact with IL-1β…” were replaced with “are required for IL-1β-induced…”. We have corrected them in this version.
  3. This revised manuscript has been edited by native English speaker.

Reviewer 2 Report

The manuscript: “Advancements in Activating Transcription Factor 5 Function in 2 Regulating Cell Stress and Survival” is well prepared review based on the the latest literature reports. The subjects of this paper is very interesting for readers of IJMS. In the paper the Authors described the biological function of ATF5, its role in physiological (apoptosis, autophagy) and pathological processes (cancer, inflammation, insulin resistance). It confirm that ATF5 is an important factor with therapeutic potential, which could be useful for diagnostics in the future. Manuscript is well structured, clearly describing the topic being addressed. Additionally the figures included into manuscript make it easy to follow for it.  

From the reviewer's duty, I would like to make a few comments:

1.       The manuscript includes the typing and syntax errors that should be corrected (e.g. page 1, line 23-25: “Activating transcription factor 5 is part of the activating transcription factor/cyclic 23 AMP response element binding protein (ATF/CREB) family of the basic region leucine 24 zipper (bZIP) transcription factors that…” ; page 3, line 112-114: “The ubiquitin proteasome degradation of ATF5 could also be downregulated by cadmium and NLK  phosphorylation, increasing the ATF5 protein” – it is not clear if it causes increase in ATF5 synthesis or increase in ATF5 transcription. This manuscript should be check by native speaker.

2.       The name of software used to prepare the figures (Fig 1-4) should be moved to other place of manuscript. The number of licence of software (IBS, Biorender) for figures preparation should be mentioned into manuscript.

3.       The title of Figure 3 should be corrected because “survival pathways” is not accurate statement because of apoptosis pathway showing as a part of this figure.

Author Response

Response to reviewers

REVIEWER2# comments

The manuscript: “Advancements in Activating Transcription Factor 5 Function in 2 Regulating Cell Stress and Survival” is well prepared review based on the the latest literature reports. The subjects of this paper is very interesting for readers of IJMS. In the paper the Authors described the biological function of ATF5, its role in physiological (apoptosis, autophagy) and pathological processes (cancer, inflammation, insulin resistance). It confirm that ATF5 is an important factor with therapeutic potential, which could be useful for diagnostics in the future. Manuscript is well structured, clearly describing the topic being addressed. Additionally the figures included into manuscript make it easy to follow for it.  

From the reviewer's duty, I would like to make a few comments:

1.The manuscript includes the typing and syntax errors that should be corrected (e.g. page 1, line 23-25: “Activating transcription factor 5 is part of the activating transcription factor/cyclic 23 AMP response element binding protein (ATF/CREB) family of the basic region leucine 24 zipper (bZIP) transcription factors that…” ; page 3, line 112-114: “The ubiquitin proteasome degradation of ATF5 could also be downregulated by cadmium and NLK  phosphorylation, increasing the ATF5 protein” – it is not clear if it causes increase in ATF5 synthesis or increase in ATF5 transcription. This manuscript should be check by native speaker.

Thank you for figuring out the problems, page 1, line 23-25 is a typing and syntax error and we have checked and corrected it. The second error on page 3 lines 112-114 is the unclear statement we wrote, cadmium and NLK inhibit the proteasome-degradation of ATF5 and stabilize the protein but does not increase synthesis or transcription, we have corrected the statement.

2.The name of software used to prepare the figures (Fig 1-4) should be moved to other place of manuscript. The number of licence of software (IBS, Biorender) for figures preparation should be mentioned into manuscript.

Thank you for your recommendation, we've moved all the software name and license number in acknowledgments in the new version of manuscript. Among them, Figure 1 only needs to cite the website without license number.

3.The title of Figure 3 should be corrected because “survival pathways” is not accurate statement because of apoptosis pathway showing as a part of this figure.

     Thank you for your suggestion, the title has been changed to “Pro- and anti-survival pathways during cellular stress and inflammation”.

Other changes:

  1. We checked spelling of author’s names and corrected two typos in this version of manuscript.
  2. In page 3, line 97 “N-terminal amino acids 1-20 of ATF5” should be “N-terminal amino acids 1-21 of ATF5”, and “directly interact with IL-1β…” were replaced with “are required for IL-1β-induced…”. We have corrected them in this version.
  3. This revised manuscript has been edited by native English speaker.

This manuscript is a resubmission of an earlier submission. The following is a list of the peer review reports and author responses from that submission.